# A New Voice of German Nationalism: An Analysis of Friedrich Schleiermacher's Nationalist Expression, 1799–1813

**William Stewart Skiles**

General Education Department, College of Arts and Sciences, Regent University, Virginia Beach, VA 23464, USA; wskiles@regent.edu

**Abstract:** Friedrich Schleiermacher emerged as a prominent advocate for German nationalism in the Wars of Liberation from Napoleonic domination in the early nineteenth century. Alongside his work as a pastor and theologian in Berlin, and also as the co-founder of the University of Berlin, Schleiermacher developed German nationalism from a distinctly Protestant perspective, one that sought the preservation of Protestantism in the German lands under Napoleonic domination. While it would be an overstatement to suggest that Schleiermacher's nationalism was anti-Catholic, he certainly sought to preserve and maintain Protestantism in the German lands. His vision of the German nation-state emphasized Germany's role as a divine instrument of God's will in the world. He assumed the cultural unity of the peoples in German lands, that is, an area of land in which the peoples are bound together by the common use of the German language. In his distinct role as a pastor, theologian, and academic, he was devoted to the cultivation of German national consciousness and the establishment of a German nation-state. Schleiermacher's work would help to provide a cultural foundation for the emergence of the German nation-state more than half a century before the establishment of the German empire.

**Keywords:** Friedrich Schleiermacher; German nationalism; Wars of Liberation; University of Berlin





## 1. A New Voice of German Nationalism

By the fall of 1806, Napoleon's army had invaded and conquered the dominant political powers in Europe. He abolished the Holy Roman Empire and handily defeated the Prussian army. It would be over six long years of occupation before Prussia would join in an alliance with its European neighbors, Austria, Russia, Britain, and Sweden, and finally expel the French from their capital city, Berlin.

On Sunday, 28 March 1813, Pastor Friedrich Schleiermacher interrupted his Lent meditations by delivering a sermon to his congregation at Trinity Church in Berlin entitled, "A Nation's Duty in a War for Freedom," which addressed the long-awaited flight of French troops out of Berlin and the welcomed arrival of Russian and Prussian troops. After expressing his fervent hope for the future of Germany's freedom, and advising his congregation on how to act in support of their Church and people, he concluded with the following prayer:

> Merciful God and Lord! Thou hast done great things for us in calling our fatherland to fight for a free and honorable existence, in which we may be able to advance Thy work. Grant us in addition, safety and grace... Grant wisdom and strength to the commanders, courage to the soldiers, faithful steadfastness to all. And grant also, as Thou canst change and turn the fortune of war, that its blessings may not be lost to us; that each one may be purified and grow in the inner man; that each may do what he can, be it much or little; that we may grow stronger in confidence in Thee, and in obedience to Thy will, an obedience reaching even to death, like the obedience of Thy Son (Schleiermacher 2004, p. 82).

Schleiermacher held such fervent nationalist sentiments on that Sunday in March that he inspired his congregation to take upon themselves what he understood to be God's work for them as a people rather than focus on a meditation on Lent or the upcoming Easter holiday. He encouraged his congregation to sacrifice even unto death—Jesus as their example—for the freedom of the German fatherland. He proclaimed that the Church was an instrument of divine grace in a world entrenched in conflict, and this was why, I argue, he could so unashamedly preach nationalism from the pulpit. He implored his congregation to accept the call of God and fight for the freedom of the fatherland, to sacrifice everything that they might develop their national character unimpeded by foreign occupiers (Dawson 1966, p. 99).

In the early nineteenth-century Protestantism and German nationalism converged in the work of Friedrich Schleiermacher. He became a renowned theologian and preacher when pietism and romanticism awakened a national consciousness based on emotion and a sense of community, and as the Wars of Liberation stoked the fires of nationalism. In Berlin, the center of Prussian politics and culture, Schleiermacher became an advocate of a new German nation founded on Protestant principles. My thesis is that he was one of the first German nationalists to advocate a nationalist vision from a distinctly Protestant perspective, arguing the unique position that Germany must be an autonomous nation-state (out from under the domination of Napoleonic France), as he stated in the prayer of 28 March 1813, to advance God's work as an instrument of divine grace in the world. Furthermore, this is a claim that he bases in part on Scripture. In making this argument he significantly contributed to establishing an inextricable link in the early nineteenth century between Protestantism and nationalism in the German states, long before the unification of the German Empire in 1871. He assumed the cultural unity of German speakers in German lands, and he often presented his vision of a united German people in contradistinction to other peoples, such as the French and the English. Schleiermacher worked toward the development of the national consciousness and the establishment of a German nation—not as a politician, nobleman, or artist, but as a clergyman and professor devoted to doing what he understood to be the work of God.

A robust historiography has emerged on the significance and impact of Schleiermacher's extensive work as a pastor, theologian, and university administrator.[1] His work has continually sparked discussion and debate. Even in his lifetime, he was understood to be forging a new path forward in the Christian faith, specifically regarding his method of prioritizing religious consciousness over dogma or Scripture (Wyman 2024, p. 289). It is for this reason that by the twentieth century, he became known as the "father" of "liberal theology" or "Protestant liberalism," though he never applied the term "liberal" to his methodology, nor did those who followed his approach, such as Richard Rothe and Albrecht Ritschl (von Scheliha 2024, p. 522). Schleiermacher's work enabled the revision of tradition so that the Christian faith aligned with developments in the natural sciences, historical and biblical scholarship, and the cultural context (Wyman 2024, p. 289). Thus, his approach broke new ground and facilitated the adaptation of the faith to the modern era.

Yet some scholars began to re-evaluate Schleiermacher's impact after the First World War as they sought to understand the catastrophic breakdown of European civilization, perceiving in his approach a subjectivism that undermined tradition. Scholars of neo-orthodox or dialectical perspectives such as Karl Barth and Emil Brunner interpreted Schleiermacher's work to be significant in contributing to the unmooring of European society from traditional Christian orthodoxy, which in time culminated in Europe's descent into a calamitous world war. They perceived in Schleiermacher's approach a subservience to Enlightenment and German Romantic conventions, sidelining Scripture and dogma, even diminishing the divinity of Christ and the doctrine of the Trinity (Jones 2024, p. 540).

However, after the Second World War and the publication of new critical editions of Schleiermacher's work, renewed interest in his contributions to hermeneutics, ethics, and pedagogy, among other fields intensified (Dole et al. 2024, p. xxv). Particular interest has emerged regarding his concern to address the cultural challenges of his day from a

decidedly Christian perspective. As Jones contends, "he had a vital grasp of Christian life in community and his thought allows for constructive conversation with diverse liberationist perspectives" (Jones 2024, p. 541). Schleiermacher's body of work continues to inspire vigorous scholarly discussion and debate, including his views on nationalism.

While scholars had been engaging with Schleiermacher's theological work even in his own lifetime and to the present day, acknowledgment of his impact on German nationalism took time to develop. (Dawson 1966, pp. 7–10). Wilhelm Dilthey was among the first to consider Schleiermacher's contribution to German nationalism. He produced an article entitled "*Schleiermachers politische Gesinnung und Wirksamkeit*" in 1862, which focused on the importance of romanticism in Schleiermacher's thought. Yet, in the following decades, not much attention was paid to his contributions to the development of the German nation-state, even after the establishment of the German empire in 1871. But at the turn of the twentieth century, renewed interest began with the theologian Johannes Bauer's *Schleiermacher als politischer Prediger* (1908), which focused on Schleiermacher's preaching activities during the Wars of Liberation. The work demonstrated that Schleiermacher was not just active in the church but a man who had a tremendous political impact in the public square. After the First World War, Paul Kluckhohn published *Persönlichkeit und Gemeinschaft: Studien zur Staatsauffassung der Deutscher Romantik*, demonstrating that Schleiermacher evolved from a romantic to a nationalist in his process of cultivating a philosophy of life. More typically in the historiography of the twentieth century, as Dawson argues, Schleiermacher is given only brief mention for his political influences on German nationalism (Dawson 1966, p. 9). Dawson addressed this lack of attention in his influential *Friedrich Schleiermacher: The Evolution of a Nationalist* (1966), which argued that Schleiermacher was influenced by pietism, rationalism, and romanticism and that he evolved from a Prussian patriot to a German nationalist. Dawson's well-researched and thorough work has been unsurpassed.

In arguing my thesis, I will first examine the intellectual and cultural context in which Schleiermacher expressed his distinctive brand of German nationalism, focusing on pietism and romanticism as significant movements that contributed greatly to the development of nationalist sentiment. I will also explore the political and social context in which Schleiermacher actually expressed a great deal of his nationalist vision, that is, after Napoleon's invasion of the German states and the ensuing Wars of Liberation. I will then analyze key texts written by Schleiermacher that provide insights into the nature of his nationalist vision. My analysis will address how Schleiermacher expressed his views on nationalism, explore the distinctiveness of his nationalist sentiment in terms of his Protestant perspective, examine what made his message effective, and lastly, offer observations and conclusions about his unique views of nationalism and propose additional avenues of research.

I have selected a range of texts to examine that demonstrate Schleiermacher's broad interests as a German intellectual: *On Religion: Speeches to its Cultured Despisers*, his classic work that explores the nature and contributions of religion in the modern age; *Occasional Thoughts on Universities in the German Sense, with an Appendix Regarding a University Soon to be Established*, his contribution as co-founder of the University of Berlin to the debate about higher education reforms in Prussia and throughout the German states; and lastly, "A Nation's Duty in a War for Freedom," a sermon delivered in the spring of 1813 to his Berlin congregation in celebration of the retreat of French forces out of the city. I have chosen these particular texts for several reasons. First, Schleiermacher expressed his nationalism in a unique way in each of these texts; he did not limit his nationalist sentiments to only one literary form. These works represent different genres, including popular theology, the political pamphlet, and the sermon. Second, Schleiermacher addressed each of these works to the German national community; even his sermons, delivered before a specific audience in Berlin, were published for popular consumption across the German states. And third, Schleiermacher expressed his nationalism in these texts in a Protestant context, relying on Protestant ideals to convey his message. These texts shed light on the nature of Schleiermacher's nationalist ideology and the manner in which he expressed it.

## 2. Historical Background

Friedrich Ernst Schleiermacher was born on 21 November 1768, in Breslau, Silesia, to devout Reformed parents who cultivated in him a love of learning and a passion for piety.[2] In his upbringing and education, his parents introduced young Friedrich to pietism, an inter-confessional religious movement that began in 1675 by the preacher Philipp Jakob Spener with his work, *Pia Desidera oder Wahren evangelischen Kirche* (Pinson 1968, p. 15). Sharing similarities with Quakerism and Methodism in England and Jansenism in France, pietism emphasized emotional enthusiasm for the gospel message; a focus on practical Christianity and the development of a personal spirituality rather than a concentration on matters of dogma; and an appreciation of the Lutheran notion of the priesthood of all believers (Pinson 1968, pp. 15–17; Greenfeld 1992, pp. 314–15). The movement breathed new life into the Lutheran and, to a lesser extent, the Reformed confessions, encouraging men and women not simply to intellectually assent to a religious dogma, but to engage in spiritual development through exercises of devotion, such as prayer and contemplation, as well as service to God through good works in the community.

According to its advocates, pietism addressed the cold rationalism and stifled spiritual growth that characterized the segments of the German churches since the advent of the Enlightenment in the eighteenth century. As Liah Greenfeld comments on the effect of pietism in the Christian community: "The de-emphasis of dogma produced a pluralistic, individualized view of religion: it was the attitude of faith, rather than its content, that mattered, and so long as one believed in Christ, it was of little consequence what else and what exactly one believed" (Greenfeld 1992, p. 318). The devastation of the Thirty Years' War in the seventeenth century was still fresh in the memories of Christians throughout the German states, and men and women expressed their spirituality with a renewed sense of freedom and courage.

Pietists advocated the priesthood of all believers, uniting believers in a common standing, claiming that each individual could relate to God through Christ in his or her own way, each in their own language and manner, without the limitation or qualification of Church dogma. This freedom of expression enabled many to enjoy a new level of intimacy with God that they may never have experienced before. The movement flourished in the German states in the late seventeenth and eighteenth centuries because it appealed to the lower classes: it fostered self-respect and prestige through spiritual edification; it provided the tools to spiritually manage a life of uncertainty, hardship, and devastation; and it encouraged their use of the vernacular for education and spiritual devotion (Pinson 1968, p. 26; Greenfeld 1992, p. 315). The movement thus served to unify the lower classes and contributed to an increased sense of common interests.

Central to the pietist movement is the concept of *Wiedergeburt*, a spiritual rebirth that distinguishes "a true believer from all others" (Becker 1991, p. 148). This was a concept that could also be applied in a nationalist context. A spiritual rebirth may best be described as a specific moment of spiritual awakening, similar to the Apostle Paul's Damascus Road experience, rather than a slow, gradual spiritual awakening over a period of many years. Conversion is the crucial moment in the life of a pietist, a defining moment that would be reflected upon for an entire life. In the early nineteenth century, particularly during and after the War of Liberation, this conception of personal, individual rebirth translated into nationalist terms: the nation may be regenerated or reborn, but this change takes an act of will on the part of the people (Pinson 1968, p. 48).

The pietist movement is central to an understanding of Schleiermacher's nationalism. It instilled in him a profound sense of the individual's dependence upon God and the importance of community (Dawson 1966, pp. 15–16).[3] But more than this, he continued throughout his life to take emotional experience seriously as a guide to approaching the challenges of life and ethical problems, not just for the individual but also for the community. He relied not only upon reason and historical evidence to understand the world but also upon the irrational or subjective, emotions and motivations of human beings. This subjective approach enabled Schleiermacher to develop his conception of nationalism

in the wake of French occupation (Dawson 1966, p. 16). The emotionalism of pietism translated quite effectively into the emotionalism of nationalism.

Romanticism was also a key element in the formation of Schleiermacher's nationalism. This movement started in the German states with the *Sturm und Drang*, or Storm and Stress, writers of the 1770s—Goethe and Schiller being the most famous. Though the early romanticists were "enlightened," the movement was a challenge to the *Aufklärung* and its stress on reason.[4] Blackbourn notes, it "looked to break through the confines of desiccated, well-tempered reasonableness (they called it mediocrity) in the name of individual genius, inspiration and feeling" (Blackbourn 2003, p. 28). The movement challenged the rationalism dominant in the age, yet still furthered the aims of the German Enlightenment's concern with understanding the human heart. Unlike pietism, romanticism did not emphasize an intimacy with God, or even a belief in God, yet both movements were concerned with exploring the emotions to develop the individual spirit and to gain a clearer perception of the world and one's place in it.

In the wake of the Napoleonic Wars, two main strands of nationalism emerged in the German states. Romantic thinkers such as Johann Gottfried Herder advanced German nationalism based on ethno-linguistic unity, cultural identity, and self-determination (Smith 2010, p. 24). Symbols of German greatness, such as monuments of victory over Napoleon's armies, were used to instill national consciousness and provide a means of expression of the German spirit (Mosse 1975, pp. 33–36). The second main strand was liberal nationalism, the adherents of which championed constitutionalism and popular sovereignty, focusing on achieving civic reforms such as democratic institutions, representation in government, and advances in individual freedoms and rights (Verheyen 1999, p. 7). As will be argued in this article, Schleiermacher's brand of nationalism was based on Romanticism, not liberalism. His writings do not include demands for a constitution or representative government but rather struggle to reveal the cultural unity of the German people as a solid basis upon which to build a nation-state, which need not be a liberal democracy. Also, one must consider the emotions often associated with German nationalism in the context of the Napoleonic Wars. Christopher Goodwin argues that "studies of boisterous 'positive' nationalism espousing national, cultural, or racial supremacy are incomplete without serious consideration of 'negative' aspects, namely fears grounded in the very real belief that a threat may extinguish a state, culture, or people" (Goodwin 2022, p. 20). Thus, the trauma of the Napoleonic Wars must be understood as elemental to German nationalism as it unified a people to prevent the trauma's recurrence (Goodwin 2022, p. 20).

Schleiermacher was introduced to the romanticists in Berlin in the mid-1790s when he became friends with the salonière Henrietta Herz, the wife of a Berlin doctor (Dawson 1966, p. 23; Hertz 2005, pp. 124–25). In her salon, Schleiermacher, who had in 1796 been appointed as pastor at the Charity Hospital, met and mingled with romanticist writers who revealed to him a profound appreciation for the German homeland, an appreciation that he had not gleaned in his theological or philosophical studies (Dawson 1966, p. 23). In this circle of romanticists, he met Novalis and the brothers Wilhelm and Friedrich Schlegel, three thinkers among others who inspired him to work out his concept of nationalism and his understanding of the relationship between the Church and the state. These ideas are evident in two important works that he published in the late 1790s, *On Religion: Speeches to its Cultured Despisers* and the *Soliloquies*. Furthermore, during this period he developed a certain nationalist pride, even chauvinism, in regard to the arts and religion in the German states; he argued in *On Religion* that Germans were a unique people, free and uninhibited to develop their innate skills and talents, particularly in the field of theology (Dawson 1966, p. 31). He praised German spirituality, comparing it favorably against that of the supposedly underdeveloped or corrupted French and English peoples.

In true romanticist fashion, Schleiermacher contributed significantly to a reformulation of what "true" religion meant, allowing religious sentiment to shape nationalist views. He argued in *On Religion* and the *Soliloquies* that "true" religion was not to be experienced in the rituals of the Church or in pietistic emotionalism, not in a knowledge or science

of God, but only in the finite human being's feeling of the infinite God in the world all around him (Dawson 1966, pp. 29–30). As Peterson contends, this "God-consciousness" is the same as the "consciousness of God," and it is based upon one's sense of "absolute dependence" upon God (Pederson 2024, p. 273). Echoing the pietist emphasis on the individual and on emotional experience, romanticism helped to shape a unique and timely religious perspective in Schleiermacher, one that inspired nationalist sentiment in a period of European crisis.

The Napoleonic conquest and occupation of the German states in 1806 was the catalyst that sparked the maturation of Schleiermacher's nationalism (Dawson 1966, p. 90). But even before then, the events in revolutionary France inspired a thoughtful reaction among the peoples of central Europe. The French Revolution of 1789 was initially supported by many in the German states—including Schleiermacher, who defended the people's right to express their national consciousness, though he did not believe Germans should follow suit and revolt (Dawson 1966, p. 90). From the example of the French Revolution, he came to understand that emotion plays a far greater role in resolving national problems than does reason; the same might not be said for the resolution of social ills. He also observed that nationalists in Prussia for some reason did not always support the nationalists in France. He questioned why not; after all, both groups want the same thing for their people—a nation of their own. Schleiermacher argued that a Prussian nationalist need not oppose a French nationalist simply because he supports the creation of separate nations—neighboring nations need not be enemies (Dawson 1966, p. 90).

Yet as the revolution turned to terror, France's neighboring states looked with unease at the possibility of revolution at home. In 1799 Napoleon Bonaparte staged his coup d'etat, took control as the Consul of France, and a few years later in December 1804, crowned himself and his wife Josephine as the French Emperor and Empress.[5] By June 1806, France had subjugated its enemies, including Austria, Russia, and Britain, and it had abolished the Holy Roman Empire. By September 1806, France declared war on Prussia and devastated its armies. All of Europe was at the mercy of Napoleon. French domination during this European crisis humiliated subjugated peoples and inspired nationalist sentiments among them. One in particular, was a pastor and professor newly appointed to the University of Halle—Friedrich Schleiermacher.

French troops billeted in his home, stole his belongings at will, and treated him and his countrymen as a conquered people. This humiliation was only exacerbated when the French government closed the University of Halle, thus ending his employment and limiting his source of income. Due to his reputation as a preacher and scholar, he was asked to participate in the establishment of a Prussian university, the University of Berlin, and by 1807 he was back in Berlin (Dawson 1966, p. 86).[6] He preached from the pulpits and lecterns in Berlin a new, more fervent Prussian nationalism, hoping to see his government and people expel the French and establish a nation of their own. Indeed, as Packman and Dole argue, "For Schleiermacher, Napoleon symbolized a totalizing Catholic vision of Europe that sought to eradicate Protestantism and, with it, the rich diversity of Christianity's religious and political forms" (Packman and Dole 2024, p. 468). He worked determinedly for the nationalist cause, even joining a secret society, the Charlottenburger Verein, which prepared for an insurrection against the French and the institution of a new post-occupation government (Dawson 1966, pp. 68–70). While it might be an overstatement to suggest that Schleiermacher's nationalism was anti-Catholic, it certainly sought to preserve and maintain Protestantism in the German lands. Thus, it is fair to argue that his form of nationalism was distinctly Protestant.

Despite Schleiermacher's efforts to inspire nationalism among his fellow Berliners, he grew increasingly pessimistic about the possibility that Prussians would ever overthrow French domination and build a nation (Dawson 1966, p. 75). By 1811, a year after he started working at the newly established University of Berlin, he grew sorely disappointed that the institution was not used by the Prussian state to instill nationalism in its student body. He realized that the people of Prussia lacked a sufficient national consciousness, that

they could not create a Prussian nation, and thus he believed that the only hope would be the awakening of a German national consciousness (Dawson 1966, p. 90). The War of Liberation from 1813 to 1814 gave him great hopes for a Prussian-led German unification against French domination, but neither the German armies' successes nor even the Allies' invasion of Paris in 1814 led to a German nationalist revolution. Even to his dying day, Schleiermacher did not witness nationalist ideology inspire the unification of the German states into one German nation-state.

Schleiermacher was certainly not alone in advocating German nationalism in the late eighteenth and early nineteenth centuries. As is often the case, intellectuals reflect on the meaning of the nation and the common bonds and experiences that unite a people during a period of political, cultural, and social change—such as that which occurred in the German states with Napoleon's invasion and the Wars of Liberation (Sheehan 1989, p. 371).[7] In the nationalist expression of Joseph Görres, a German journalist and propagandist, for example, we find an emphasis on the immutability of German culture and the persistence through the ages of the German *Volk*. Görres advocated the establishment of national institutions and a constitution to protect German culture and the *Volk* from the vicissitudes of war, invasion, and foreign domination. Perhaps the most influential nationalist of the period was J.G. Fichte, a renowned philosopher who advanced cultural and political Prussian nationalism. In the words of one scholar, Fichte argued that "language is the key to human identity, its purest expression is the most sublime form of collective existence" (Sheehan 1989, p. 376). He envisioned a fantastic role for the German people: they alone could save the ailing human culture because they still lived in their ancestral land, spoke their ancestral language, and conducted themselves according to their ancestral moral code. The Germans, according to Fichte, were a people with a long and sacred tradition that had the potential to advance the human race. Likewise, Ernst Moritz Arndt was an early German nationalist who believed that language was the basis of a national identity and that Germans possessed a superior culture. He was one of the few early nationalists to publish a program for German unification: a monarchical state centered in Berlin, yet with a democratic foundation. There were other nationalists, of course, including the political figure Friedrich Ludwig Jahn, who incorporated mystical and racial elements in his nationalist vision, and also the romanticist soldier and songwriter Theodor Körner. Yet, as I will argue throughout this article, Schleiermacher was unique in that he was a well-respected pastor and professor who worked within the institutions of the church and university, who advocated nationalism from a distinctively Christian perspective. Germany's purpose, he argued, was its divine calling, to work in the world as an instrument of God's grace.

An examination of his life shows that he devoted much time and energy to advancing the cause of nationalism and creating a nation-state for his people. However, contrary to Dawson, it is too much to argue that nationalism was the most important factor in his life (Dawson 1966, p. 160). Despite his preaching nationalism from the pulpit and lectern and his work with the Charlottenburger Verein during the War of Liberation, we must keep in mind that Schleiermacher was foremost a pastor and a theologian committed to studying and, even more, experiencing God. He understood his nationalism to be in the service of the Church, to be God's work. This is why he believed nationalism to be a program of peace between peoples—each cultivating a national consciousness for their own political and cultural advancement—instead of a source of division. It is thus quite an overstatement to contend that he worked tirelessly for nationalism as a good in itself without considering its spiritual implications—or the purpose to which it must be put, i.e., the work of God.

I have attempted to provide a context in which Schleiermacher's nationalism developed. Pietism and romanticism encouraged him to investigate the realm of emotions to understand the world, while at the same time emphasizing the centrality of the individual in the context of a highly valued community. The French Revolution, Napoleon's invasion of the German states, and the subsequent War of Liberation provided Schleiermacher with a turbulent European political and social context in which to think about the nature of national consciousness and the possibilities of nation-building. Though we cannot conclude

that Schleiermacher would never have developed nationalist sentiments without these historical realities, we may nevertheless contend that his nationalism would not have been nearly the same.

### 3. Expressions of a Nationalist

Nationalism is a prominent theme in one of Schleiermacher's most popular works, *On Religion: Speeches to its Cultured Despisers*, published in 1799. The book at once established his reputation as a Protestant leader in the German states. It is a compilation of five speeches of popular theology written to the "cultured despisers" of religion, or those who thought religion could make no contribution or had little value in the modern world. It is a confession written to the intellectuals of his day, specifically "the literary and philosophically oriented," and thus, it was not written in the style of academic theology but in the manner of romantic literature (Redeker 1973, p. 35). Schleiermacher argued that, "The sum total of religion is to feel that, in its highest unity, all that moves us in feeling is one; to feel that aught single and particular is only possible by means of this unity; to feel, that is to say, that our being and living is a being and living in and through God" (Schleiermacher 1994, pp. 49–50). Schleiermacher's pietist and romanticist background is clearly evident in the emphasis on feeling and experiencing God through the world itself, rather than on proving or explaining dogma.

The main purpose of *On Religion* is to advance a certain perspective of religion as a positive and even needed component in human life, and thus any nationalist sentiments found in the work must be understood in this context. In the first speech, Schleiermacher expressed a sense of German pride meant to encourage in his readers a desire to explore the issues of spirituality. He wrote,

> If I am thus impelled to speak of religion and to deliver my testimony, to whom should I turn if not to the sons of Germany? Where else is an audience for my speech? It is not blind predilection for my native soil or for my fellows in government and language, that makes me speak thus, but the deep conviction that you alone are capable, as well as worthy, of having awakened in you the sense for holy and divine things (Schleiermacher 1994, p. 9).

Schleiermacher addressed his speeches not to the "cultured despisers" of religion throughout all of Europe, but only to those in "Germany" because he believed they alone are "capable" and "worthy" of appreciating what he had to say. Note that Schleiermacher defined "Germany" as his "native soil," and home to his "fellows in government and language." His definition of "Germany" is bound by the German soil, German governance, and German language. This quotation might at first glance appear to be empty flattery—that only Germans are capable of understanding his argument—but then he criticizes "those proud Islanders" (the English) for their "zeal for knowledge" and "worldly wisdom," because they most certainly cannot appreciate his emphasis on feeling and its importance in spiritual expression (Schleiermacher 1994, pp. 9–10). And he also criticized the French because "they tread [religion's] holiest ordinances under foot," a likely reference to the destructive dechristianization campaigns that occurred throughout revolutionary France in the past decade (Schleiermacher 1994, p. 10).[8] Early in Schleiermacher's speeches, he appealed to his fellow Germans as a spiritually sophisticated people in contradistinction to the English and the French, who, he thought, could not possibly understand or appreciate what he had to say. This is meant no doubt to flatter his readers and to inspire a sense of national pride, which here occurs at the expense of other people groups.

Schleiermacher acknowledged the problem of spiritual unity in the German states, divided as the states were among Protestants—the Lutheran and the Reformed confessions—and the Catholics. This is particularly evident in the fourth speech. If the Church is divided, according to Schleiermacher's reasoning, it could not as effectively serve society or the interests of the state. In this sense, it is in the state's best interest to support a unified Church (Schleiermacher 1994, pp. 176, 207). As early as 1799, he seemed to advocate for the union of the Protestant churches into an Evangelical Church, which he hoped

would invigorate religious devotion among Germans and promote a greater degree of cultural unity (Schleiermacher 1994, pp. 176, 207). Such a union may resolve the tensions between Protestants, but Schleiermacher provided no solution to bringing Protestants and Catholics together. In fact, he admitted to "almost always [speaking] as if all Germany were Protestant"; this is due in no small part to his church service in Prussia and the northern and central German states (Schleiermacher 1994, p. 274). He clearly recognized that religious disunity was a great problem among the German people, a substantial impediment to national unity, and yet by 1799, he could find no workable solution.

Yet Schleiermacher offered the potential for the German university to foster nationalist sentiment and work toward national unity. Schleiermacher presented ideas on German university reform and a proposal for a new Prussian university in his influential political pamphlet, *Occasional Thoughts on Universities in the German Sense, with an Appendix Regarding a University Soon to be Established*. Published in 1808, the text followed Friedrich Wilhelm Schelling's proposal of 1803 and J.G. Fichte's pamphlet of 1807, which would soon hasten the proposals of other leaders in German education, including Heinrich Steffen and Wilhelm von Humboldt. These five texts offered suggestions to reform the German university, and notably, they confronted the problem of state support of, and interference in, higher education. Schleiermacher exercised considerable influence in the development of the University of Berlin, and his pamphlet reveals him as pragmatic, accommodationist, and respectful of German educational traditions (McClelland 1980, p. 120). In *Occasional Thoughts on Universities in the German Sense*, Schleiermacher reflected on what it would take to establish a successful university—one that would draw many students and exert considerable influence in Germany—and that would serve the interests of the state. Yet at the same time, the university must remain an environment characterized by intellectual freedom and a concern for independent research. After publishing this pamphlet, Schleiermacher was chosen by the Prussian state to lead the commission in selecting the new faculty of the University of Berlin.

As in *On Religion*, Schleiermacher assumed a cultural unity of the German states. He often referred to "Germany" or the "fatherland" without explicitly stating what he meant. Central to this cultural coherence is the common German language. For instance, in examining the role of the state in relation to scientific endeavor, he believed that it was "unnatural for an area that has one and the same language to be divided into so many small states as Germany sustains." (Schleiermacher 1991, p. 6). We can surmise that when Schleiermacher wrote of "Germany" or the "fatherland," he meant the land of German speakers. As Theodore Vial contends, Schleiermacher argued that "Germany was a cultural/linguistic Volk, not defined by blood," and thus "[o]ne could be Prussian and Jewish" (Vial 2024, p. 607). Language unified the people. For Schleiermacher, the division of the German states was an impediment to the greater cooperation of universities and scientific academies throughout the German-speaking lands. In this particular context, Schleiermacher used the phrase "German fatherland" to refer to the cultural unity of the German states in terms of language. He hoped for a time when state boundaries would cease to be an impediment to scientific research, when the German-speaking people could share knowledge and assist one another without state interference.

A sense of national pride is also present in this pamphlet, even connoted in the title itself, *Occasional Thoughts on Universities in the German Sense*, which we may infer to mean in contrast to the "French sense" or the "English sense." In delineating the differences between academies as scholarly societies, schools as centers of teaching, and universities as centers of research and teaching, Schleiermacher argued that these distinctions are "originally German and follow exactly the cultural example of other relationships that have come out of Germany: the school as the being together of master and apprentices, the university as the being together of master and journeymen, and the academy as the gathering together of masters" (Schleiermacher 1991, p. 11). He argued that Germany is the model of European centers of higher education. Germany is the leader, and has been for centuries, in education in all of Europe. This comment reflects cultural pride and also the desire to look within

the German fatherland to find solutions to university reform rather than to other countries, such as France or England. But more than this, as the quotation indicates, Schleiermacher understood one significant task of the university to foster a sense of community (Anderson 2004, p. 72). This is one key outcome of the university—one way the university contributes to the common good. Student associations, including political and military organizations, were to contribute to the building of community, as well as to function as avenues for academic freedom for students, including the discussion of political ideals and how to achieve them (Anderson 2004, p. 72). Anderson contends that "there is no doubt that the liberation of Germany created a fervent desire among the younger generation [of students] for national renewal and for a new political order" (Anderson 2004, p. 72). The German model was an exemplar that demonstrated how academic, social, and cultural flourishing happens in a university.

Schleiermacher viewed the university as an institution to serve the advancement of German cultural unification for the purpose of doing God's work in the world. Drawing an analogy between the individual and the state, Schleiermacher argued:

> A noble and upstanding life is no more possible for a state than for the individual if its necessarily limited abilities in the realm of knowledge are not linked to a higher calling. Similarly, the body of knowledge itself must naturally and necessarily be grounded in systematic inquiry for the state no less than for the individual, and only through systematic inquiry can it truly be perpetuated or brought to fruition. The state thus tries to forge vital connections by every means possible (Schleiermacher 1991, p. 66).

While Schleiermacher is more explicit in speaking in religious language in his sermons (as we will see), he indicated in this text that, like the individual, the state must align itself with a "higher calling," presumably God's will, that it might bear good fruit for the benefit of the whole state. This is the "higher calling" of the university. The university must be an agent of unity, illumination, and vitality for the German people.

Schleiermacher asserted that the universities in the German states must not be so different from each other in function and philosophy that they further divide the German people, which could only work to their detriment. Central to his point is that every state should not have its own university to invest and take pride in, but rather that one magnificent university should reach out and serve many states. In this way the university would not be a source of regional pride and distinction, but a source of national unity. With this in mind, Schleiermacher argued for the establishment of the new Prussian university to be located in Berlin. He wrote, "Prussia does not want to isolate itself but wishes also in this respect to remain in vital alliance with the entire natural area of Germany" (Schleiermacher 1991, p. 66). From the center of Prussian politics and culture, a university in Berlin has the potential to influence not only all of Prussia and northern Germany but the whole of the "natural area" of Germany. One might well ask what he means by "natural area," but we can surmise that he means something similar to the German fatherland, the land united by the common use of the German language. Thus, the new University of Berlin is to be an institution designed to further unify the German nation—at the very least in a cultural sense.

*Occasional Thoughts on Universities in the German Sense* is clearly the least religious text of the three treated in this article, yet Schleiermacher's Protestant background is evident in his proposal for the University of Berlin. More specifically, his pietist background likely informed his contention that the university should impose no doctrinal requirements on professors, but rather encourage and safeguard intellectual freedom. All professors should be free to pursue their own interests without fear of administrative reprisals for having a faith different from that endorsed by the institution. Schleiermacher had first-hand experience of colleagues reprimanded at the University of Halle for falling out of step with doctrinal orthodoxy. One can make the case that Schleiermacher's perspective of intellectual freedom was at least partially informed by his pietist upbringing and its emphasis on the freedom of personal spiritual growth over strict doctrinal adherence.

As for the university, so also for the church. Schleiermacher admonished and inspired his congregation to live in a manner worthy of the freedom the Lord granted to the German people. He, like other pastors, used the sermon as a way to "[instill] the notion of a shared German national spirit, morality, and culture" in the wake of Napoleon's conquests (Ravenscroft 2023, p. 299). Five years after publishing *Occasional Thoughts on Universities in the German Sense*, Schleiermacher delivered the sermon "A Nation's Duty in a War for Freedom," the text that opened this article. As previously mentioned, he interrupted his planned Lent mediations to take the time to celebrate with his congregation at Berlin's Trinity Church the end of French occupation and the arrival of Prussian and Russian troops. The sermon is remarkable because Schleiermacher opened with an apropos reading of the Hebrew Scriptures, Jeremiah 17:5–8, and 18:7–10. He read from the prophet Jeremiah, who served his people in the last years of Judah's political independence before the Babylonian captivity in the sixth century B.C.E. In the first passage, Jeremiah called the people of Judah to repentance for apostasy, to turn and trust in the Lord before the Babylonians invaded. In the second passage, Jeremiah quotes the words of the Lord:

> At one moment I may declare concerning a nation or a kingdom, that I will pluck up and break down and destroy it, but if that nation, concerning which I have spoken, turns from its evil, I will change my mind about the disaster that I intended to bring on it. And at another moment I may declare concerning a nation or a kingdom that I will build and plant it, but if it does evil in my sight, not listening to my voice, then I will change my mind about the good that I had intended to do to it (18:7–10, NRSV).[9]

Like many preachers in their own day, Schleiermacher drew a parallel between Judah and his homeland. This text differs from the other two texts discussed thus far, predominantly because Schleiermacher's task here was explicitly to inspire nationalist sentiments in his audience; his nationalism is not expressed simply in the service of a greater project, such as the defense of religion or the proposal for a new university. Here, his message is essentially nationalist. Schleiermacher hoped to encourage his audience to fight in what he called a "holy war," a war not of conquest, but rather one in which the fatherland would become what God wills for it. Reflecting back on the text from Jeremiah, Schleiermacher pointed out the German people's national sin, its arrogance and reliance upon military might: "We became the man who makes flesh his arm and whose heart departed from the Lord" (Schleiermacher 2004, p. 70). Germans failed to trust in God, he argued, and did not repent of their self-reliance. Schleiermacher continued, "suddenly the Lord spoke out against us as against a nation and kingdom which he would pluck up and pull down and destroy. Then there fell upon us that grievous, crushing disaster in war, and this sudden fall from the height into the abyss was followed by the ever more deeply and painfully suicidal calamity of peace" (Schleiermacher 2004, p. 71). Schleiermacher appeared to believe that God humbled all Germans with the invasion of Napoleon's armies, and yet God had just granted the people a new opportunity to repent; they must unite and serve the common cause—their nation's liberation.

Schleiermacher contended that Germany had a God-given distinctive character manifested in its laws and culture, and more than this, that this is true of every people. Further, according to Schleiermacher, God has given to each people a land that is inviolable, and that must be respected by all nations. He wrote, "Every nation, my friends, which has developed to a certain height is degraded by receiving into it a foreign element, even though that may be good in itself; for God has imparted to each its own nature, and has therefore marked out bounds and limits for the inhabitants of the different races of men on the face of the earth" (Schleiermacher 2004, p. 73). This is a remarkable statement because it seems to suggest that people and nations are tied to the land they inhabit and that foreign invasion is not only a crime against other people but also against God's purposes and design for a people. Again, we see Schleiermacher's understanding of the nation as a bounded area of land occupied by a specific people group. Germany's expulsion of Napoleon's forces is thus a "holy war" in the sense that Germany must fight for God's will in ridding its land of

foreigners. In fighting this war Germany demonstrates its trust in the Lord, that it "means to defend at any price the distinctive aims and spirit which God has implanted in it, and is thus fighting for God's work" (Schleiermacher 2004, p. 74). Drawing from Jeremiah again, Schleiermacher contended that only in this way may Germany become a nation "like a tree planted by water,/sending out its roots by the stream./It shall not fear when heat comes, and its leaves shall stay green;/in the year of its drought it is not anxious,/and it does not cease to bear fruit" (17:8). Elemental to Schleiermacher's vision of a unified Germany is that it will bear good fruit—it will be an instrument of God's divine will in the world.

The proper response to God's gift of liberation is sacrifice. Schleiermacher encouraged each member of his congregation to sacrifice in his or her own way for the defense of the "German fatherland." His sermon meant not only to inspire religious feeling but to encourage his congregation to action. Schleiermacher read from the pulpit King Friedrich Wilhelm's summons to the *Landwehr*, a civilian military force of males between ages seventeen and forty, meant to supplement the strength of the regular army (Schleiermacher 2004, p. 76; Dawson 1966, pp. 104–6). In the *Landwehr* men of all classes and stations in life may stand together to fight for their fatherland. Schleiermacher proclaimed, "What an exalted feeling this call must awaken in all of us! What a firm confidence in the strength thus united! What a happy foretaste of the harmony and love in which all ranks will be bound together, when they have all stood side by side face to face with death for the Fatherland!" (Schleiermacher 2004, p. 76). Schleiermacher realized that he must not only call sons and fathers to sacrifice for their nation but also wives, mothers, friends, and employers. All must sacrifice, and he calls the members of his congregation to do what they can in defense of the fatherland.

Likewise, thanksgiving is a necessary response to God's gift of liberation. Schleiermacher offered a prayer thanking God for calling "our fatherland to fight for a free and honorable existence, in which we may be able to advance Thy work" (Schleiermacher 2004, p. 82). Once again, God's work is revealed as the central theme at the foundation of Schleiermacher's nationalism. The fatherland exists not for its own purposes, but to do the will of God in the world. This is a crucial point for Schleiermacher. In this sermon, Schleiermacher encouraged his congregation to do what he thought God demanded of the German people, and that is to fight for its "free and honorable existence." He called for the obedience of his congregation to do the will of God, and in the final words of the sermon asked them to obey even if it meant death, with Jesus as their example. In this sermon, Schleiermacher placed the nation's existence and well-being at the heart of Christian service to God.

The theme of the fatherland in service to God in this sermon is evident in his other sermons of the period as well. Perhaps the most clear and explicit example is a sermon entitled "Rejoicing Before God," delivered on the anniversary of the Battle of Leipzig, on 18 October 1818. In this sermon, Schleiermacher gives thanks and praise to God for his guiding hand in shaping the history of the fatherland, going so far in the concluding prayer to invoke God's special blessing on the nation:

> Thou didst raise us up when we were crushed and had almost perished! It is Thy will to make us again a vessel unto honour, after we were despised and seemed like a vessel of wrath; Thou alone hast done it, to Thee let all our hearts be devoted. Rule Thou in our hearts as Thou has outwardly ruled among us. Make us by Thy Spirit more and more a people to Thy praise, *a royal and priestly nation*; govern us by Thy word and Spirit that we may be ever becoming worthier of that highest name which we bear, the name that comes to us from thy Son. Amen. [Emphasis added] (Schleiermacher 2004, p. 194).

By petitioning God that the German people may be "a royal and priestly nation," Schleiermacher purposefully invoked God's redemptive history—using human beings, from Adam and Eve, through the nation of Israel, and to the Christian Church—to proclaim the love and justice of God to all the world, to give the praises of all creation to God, and to be good stewards of God's creation. In short, "a royal and priestly nation" is to show the nations of

the world who God is and how to rightly relate to him, as a people that care for and have dominion over their corner of the world.[10] Schleiermacher called the German nation to a high standard of service to God and his kingdom.

Schleiermacher perceived Germany as a force for good in the world. Another example will suffice to support this point. Schleiermacher preached a sermon on Trinity Sunday in an undisclosed year, but published in 1814, entitled "Necessity of the New Birth," and it was based upon the story of Nicodemus in John 3:1–8, in which Jesus tells the pharisee that "unless one is born again, he cannot see the kingdom of God" (NKJV). Schleiermacher preached, as was and is typical, on the need for transformation of the individual by the work of the Holy Spirit for newness of life. But he connected this necessity with national regeneration. He preached, "And so for every nation the appearing of the gospel in it is its regeneration, not only a perfecting of its 'former condition'; for, as we learn from history, much that was really good and beautiful often perishes in the first place, and the whole form is changed, the whole life takes another direction" (Schleiermacher 2004, p. 90). He then returned to a discussion of individual spiritual rebirth. The fact that Schleiermacher couched his discussion of national rebirth with individual rebirth indicates he perceived the two types of rebirth as analogous: the dying of the old, the birth of the new, and the emergent fruits of the newness of life that contribute to the work of God. Schleiermacher admonished his listeners that this consciousness of rebirth ought to be their "earnest endeavor to help others forward into this new life," that "even we may help in the work of the Spirit of God" (Schleiermacher 2004, p. 102). National regeneration should mirror individual rebirth, complete with newness of life, but also the bearing of good fruit that brings glory to God.

It is difficult to say for certain how the German public received Schleiermacher's works, but here I offer a few comments that might prove helpful. *On Religion* became an instant success among the German Protestant public, establishing his fame as a leading pastor and theologian. In fact, this work led the administration of the University of Halle to hire him as a professor of theology in 1800 (Schleiermacher 1994, pp. xi–xii). Public demand allowed Schleiermacher to release two updated editions of the book in 1806 and 1821, further evidence of the book's lasting resonance among German Protestants. Since then, *On Religion* has become a theological classic in the Protestant liberal tradition; it has been in print ever since, even in English. Schleiermacher's sermon, "A Nation's Duty in a War of Freedom," was preached on 28 March 1813, and was included in his third volume of collected sermons. When reading Schleiermacher's sermons, one must keep in mind that he preached from notes, not a text, and that he only later wrote them out in full and prepared them for publication. The genre of the sermon is meant to be spoken and heard, not read, and therefore, when reading Schleiermacher's sermons, we necessarily miss the inflections and intonations that reportedly made him a superb preacher. Both this sermon and *On Religion* were meant for the general reading public of Protestant Prussia, and perhaps even all of the German states. It is likely that theologians and the laity read and appreciated these works, given their popular style and subject matter. His political pamphlet *Occasional Thoughts on Universities in the German Sense* is a very different work. Schleiermacher does not explicitly state his intended audience in the text, yet we can infer from the subject matter that civil servants and educators read this work rather than the general reading public. One may conclude that this work was indeed influential because, as previously noted, the Prussian state appointed Schleiermacher to help found the new University of Berlin based upon it.

## 4. Observations and Conclusions

Modern readers must be careful not to read into German history a unity or coherence that did not yet exist. Schleiermacher's use of the term *Deutschland*, "Germany," is problematic because, as James Sheehan has noted, such a unified political, cultural, social, economic, or geographic entity did not exist in the early to mid-nineteenth century: "To suppose otherwise is to miss the essential character of the German past and the German present: its

diversity and discontinuity, richness and fragmentation, fecundity and fluidity" (Sheehan 1989, p. 1). Schleiermacher uses the term "Germany" to speak of a culturally unified people bound together by the common use of the German language and occupying an area of land, which he does not clearly define. For Schleiermacher, the common language is the basis of German identity. We cannot speculate about the geographic boundaries of this conception of Germany; we cannot determine whether he refers to the old, destroyed Holy Roman Empire, to the present system of states and confederacies, or to a wholly new vision of a future German nation. Nor should we speculate about his thoughts on how the German speakers of the Habsburg Empire fit into his conception of "Germany." The evidence from the primary sources examined in this article supports the conclusion that Schleiermacher used the term to refer to a cultural unity of the German people, though he did not define or elaborate on this unity—he merely assumed it.

A second conclusion is that Schleiermacher's nationalist sentiment almost always occurs in the context of a problem or an issue relating to the French or English peoples. In *On Religion*, Schleiermacher expressed a national chauvinism in singling out as his audience the more spiritually sophisticated Germans, as opposed to the irreligious French and the worldly-wise English. Written just two years after Napoleon's invasion, occupation, and administrative reforms, Schleiermacher's *Occasional Thoughts* articulates his beliefs about universities in the "German sense," that Germans must treasure their long and admirable tradition of education. Finally, Schleiermacher's sermon, "A Nation's Duty in a War for Freedom," is overtly nationalistic and antagonistic to the French, particularly in his call to his congregation to take up arms and defend their lands. The German is measured and defined against the French, "the other." The German people are unified as a group in contradistinction to the French: their integrity is compromised by the French invasion and occupation; their spirit is broken by the humiliation caused by the French; and their survival necessitates the defeat and expulsion of the French.

Third, Schleiermacher expressed his nationalism often in the context of an institution. He delivered his sermons in Berlin's Trinity Church, an institution with a particular context, history, and role in the community. Though we certainly cannot conclude that all his parishioners shared his nationalist sentiments, or appreciated them expounded from the pulpit, we can surmise that his words were given sanction by the traditional Protestant authority of the pulpit and the church, far more so than if he gave this same sermon on a street corner. In a different way, Schleiermacher sought to change the institution of the German university, arguing his proposed reforms from a nationalist perspective. We should remember that he was a pastor and professor, a man who functioned in two very important social and cultural institutions: the church and the university. We may conclude, at the very least, that Schleiermacher articulated his nationalism within the context of institutions, and thus, nationalism had an effect on these institutions. To measure and determine this effect would proceed too far afield from the topic of this article.

Schleiermacher also most often discussed his nationalist sentiments from a distinctly Christian perspective. He was a Protestant pastor and professor who used tools from his Reformed tradition to express his viewpoints. For example, the sermon, "A Nation's Duty in a War for Freedom," does not simply inspire nationalism or call his congregation to take up arms in defense of their nation, but rather anchors these aims in the demands of Scripture; thus, Schleiermacher followed the Reformed tradition of the "scripture principal," the method that ensures that preachers speak the word of God rather than the word of man (Barth 1993, p. 526).[11] *On Religion* is a book that highlights the wide range of beliefs acceptable in the Protestant tradition; the important element in faith for Schleiermacher was one's utter dependence upon God, the God who reveals himself all around us. This work is free from the confines of Catholic dogma, and it demonstrates a unique combination of romanticist and pietist (a distinctly Protestant movement) religious expression. It may be obvious to conclude that a Protestant pastor and professor expressed his nationalism in Protestant terms, yet we should not forget that one can only speak of nationalism

in a context, and that this context in turn shapes and informs that very understanding of nationalism.

A final conclusion that we may draw is that Schleiermacher perceived the problem of religious disunity in the German states—that Germany could not be united politically or socially until the Catholic and Protestant churches were united—yet he proposed no viable solution. This dilemma is most clearly seen in the fourth speech in *On Religion*, where he contends that a Church divided could not as effectively serve society and the state as could a Church united. One unified Church could present a coherent social and cultural message to its people, train children with a single catechism, and provide consistent and single-minded answers to political problems. In 1799, in his work *On Religion*, Schleiermacher proposed a Protestant Church, a union of the Reformed and Lutheran churches. Yet he does not provide any solutions to a Catholic-Protestant union, nor does he address how to include Jews in a culturally unified Germany. Schleiermacher's acceptance of this problem of religious disunity almost contradicts his assumption that Germany is a culturally unified land; I say "almost" because he believes that the German language is the basis of German cultural unity.

This article raises additional avenues for further research. First, it may be profitable to examine the distinctively Protestant language, concepts, and methods that Schleiermacher used throughout his career to advocate nationalism. We may explore what religious and political implications this may have for the development of early German nationalism. Second, a more thorough investigation into Schleiermacher's views of German religious unity and diversity might better enable us to appreciate the challenges confronting the possibility of German political unity in the early nineteenth century. Third, his political pamphlet, *Occasional Thoughts*, raises the question about his views on the role of education in advancing nationalist ideals. An investigation of his lecture materials and any other of his works on education may help us understand what political aims he may have advanced in the classroom. And fourth, because Schleiermacher advocated his nationalist ideas from within important German institutions, we might well ask what role institutions had in facilitating the development and spread of early German nationalist ideology.

In the wake of the Napoleonic Wars, Friedrich Schleiermacher struggled for the cause of German nationalism, advocating a distinctive nationalist vision as a Protestant pastor and professor—one which emphasized Germany's role as a divine instrument of God's will in the world. His work reverberated beyond the walls of the sanctuary and the lecture hall to help provide a cultural foundation for the emergence of the German nation-state more than half a century before the establishment of the German empire.

**Funding:** This research received no external funding.

**Data Availability Statement:** All data is provided in the sources included in the References.

**Conflicts of Interest:** The author declares no conflicts of interest.

## Notes

[1]     This paragraph is based on Dole, Poe, and Vanderschel's discussion in their "General Introduction", in (Dole et al. 2024).

[2]     The Reformed tradition developed in distinction to Lutheranism in the period of Protestant division during the sixteenth century. The original split occurred between Martin Luther and the Swiss pastor Ulrich Zwingli over the issue of Christ's presence in the Eucharist; John Calvin weighed in on the dispute, offering his own interpretation. Zwingli and Calvin became the two great leaders of the reform movement, working to advance more extensive reforms in the Church in areas such as liturgy, theology, and church governance. After the Peace of Westphalia in 1648, the princes of the German states accepted either of the Protestant traditions or Roman Catholicism, and imposed it upon the people in their lands. In this way, the various confessions took root in the German political and social fabric. For more history and an elaborate discussion of the distinctiveness of Reformed Theology, see "Reformed Theology," in *New Dictionary of Theology* (Ferguson et al. 1988, pp. 569–72).

[3]     In this paragraph I am relying upon Jerry Dawson's analysis of pietism and its influence upon Schleiermacher's development as a nationalist, in (Dawson 1966, pp. 15–16).

[4]     This discussion is largely reliant upon (Blackbourn 2003, p. 28); and also (Greenfeld 1992, pp. 322–30).

[5]     For the chronology, I am relying upon Clive Emsley's *Napoleon* (Emsley 2003, pp. xi–xv).

6   The Prussian state created the University of Berlin in 1809, and its first sessions were held in 1810.

7   This discussion on early German nationalists relies upon Sheehan's account (Sheehan 1989, pp. 371–88).

8   In a note in a later speech, Schleiermacher writes that he and many others in Germany felt the French showed themselves to the world in the Revolution as "thoroughly selfish" and "vacillating" (Schleiermacher 1994, p. 255).

9   All Bible passages are in the New Revised Standard Version.

10  This language of a "royal and priestly nation" goes back to the Hebrew Scripture and deeply informs the Church's conception of its role in the world. N.T. Wright argues, "The main task of this vocation is 'image-bearing,' reflecting the Creator's wise stewardship into the world and reflecting the praises of all creation back to its maker. Those who do so are the 'royal priesthood,' the kingdom of priests,' the people who are called to stand at the dangerous but exhilarating points where heaven and earth meet." (Wright 2011, p. 76).

11  A discussion of this method and its importance in the Reformed tradition may be found in Barth (1993).

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
