# Peer review of "A New Voice of German Nationalism: An Analysis of Friedrich Schleiermacher’s Nationalist Expression, 1799–1813"

_religions, doi:10.3390/rel15060684_

Round 1
Reviewer 1 Report
Comments and Suggestions for Authors
This is a very interesting article addressing a topic that, as far as I am aware, has so far received insufficient attention in relation to the life and work of Schleiermacher. It therefore makes a welcome contribution both to studies on his life and work, but also to understandings of the interface between religious orientations and nationalisms of various kinds. I also think that, in thematic terms, it fits well within the Religions special issue to which it has been assigned. I have no particular points of substantive criticism to make or requests for further work. In my overall view, the piece is fundamentally ready to proceed to publication.
I have, however, spotted two very small textual/presentational matters which it would be useful if the author could attend to before the article is published:
(1) At line 233/34 I think that within the same sentence the reference to the "early nineteenth" century is repeated, whereas I think it likely that the author's intention had been to refer to first to the "late eighteenth" century and then to the "early nineteenth" century.
(2) At line 273, the sentence should begin with a capitalization of the word "pietism" as "Pietism".
Author Response
I thank the reviewer for their insights and helpful comments. They were truly invaluable as I've made major revisions to the article. I've added 2000 words on the historiography and the historical-theoretical aspects of German nationalism. I've also updated the thesis to reflect these changes.

Reviewer 2 Report
Comments and Suggestions for Authors
In this paper the author examines three texts by Friedrich Schleiermacher to demonstrate that he promoted German nationalism during and immediately after the Napoleonic Wars. The texts are well chosen and do indeed demonstrate Schleiermacher's commitment to the Prussian cause couched in the rhetoric of German nationalism. The paper nonetheless needs more precision in terms of what it is arguing, more attention to the historical context, and a greater claim to originality before it is ready for publication.
Originality
The argument that Schleiermacher was a German nationalist has already been made convincingly by Jerry Dawson in his 1966 biography of Schleiermacher, and it is not clear how the present study advances Dawson's findings. The section on Schleiermacher's evolution from a patriot to a nationalist (pp. 5-6) relies almost exclusively on Dawson, for example, and the analysis of the texts simply reinforces Dawson's conclusions. More recently, Ruth Jackson Ravenscroft has made the same argument in her Introduction to the Oxford History of Modern German Theology, vol. 1 (2023). The author needs to include a thorough discussion of the historiography on Schleiermacher and show how this article adds to that historiography. A good place to start might be The Oxford Handbook of Friedrich Schleiermacher (2023), paying particular attention to the chapters by Jorg Dierken, Johannes Zachhuber, Miriam Rose, Arnulf von Scheliha, Theodore Vial and Wilhelm Grab. Look at the historians that are being quoted in those chapters and engage with them.
Historical Context
More work needs to be done on how others were expressing German nationalism during this period to show whether Schleiermacher was a pioneer or was simply imitating what he was hearing around him. Because he was working in the university, discussing the nationalism of the Prussian student movement (and Schleiermacher's reaction to it!) at this time is crucial. Start with R. D. Anderson, Students, Professors and Politics (2004) and go from there. Similarly, the author needs to discuss the state of German nationalism in this period in more detail than is given in the short survey on pp. 6-7. George Mosse, The Nationalisation of the Masses (1975) is still indispensible on this topic, but there are plenty of more recent studies, including articles such as Christopher Thomas Goodwin, "Surviving Crisis", War and Society, 41 (2022).
Precision
Closer attention to the historical context of German nationalism in this period will make clear the need for more precision in terms of terminology. On p. 2, for example, the author uses "nationalist sentiments" and "patriotic hopes" interchangeably, as if nationalism and patriotism were the same thing, and on p.12 they equate "our fatherland" with "the nation". Be very careful to avoid these sorts of slippages. In particular, it is not at all clear what sort of nationalism Schleiermacher is advocating here. Was it a romantic nationalism based on culture (a la Herder) or was it a liberal nationalism that presupposed a constitutional monarchy (a la 1848)? Have a look at Karolewski and Suszycki, The Nation and Nationalism in Europe (2011) for a good introduction to the varieties of nationalism, but it is also worth looking at Rogers Brubaker, Nationalism Reframed (1996) for a more sophisticated discussion of how nationalism functions in political rhetoric of the sort Schleiermacher was promoting. Part of the problem here is the author's reliance on Dawson, who was writing in a period before this sort of precision entered into the study of nationalism. A more sophisticated understanding of the different types of nationalism might allow the author to frame the article as a critique and extension of Dawson's work.
The author also assumes that Schleiermacher thought that "Germany" meant "the lands where people speak German". This is asserted several times but never proven. It is very unlikely that this is the case, given how few German nationalists of this period included the Habsburg lands within their definition of Germany. This assumption needs to be rethought and a more robust conclusion proven rather than asserted.
The fact that Schleiermacher's nationalism was "distinctly Protestant" is an important feature of this article, but it is not clear what is particularly Protestant about it. What was his attitude towards German-speaking Catholics, for example? Did he imagine them to be part of the German nation? The suggestion that his commitment to intellectual freedom came from his Pietism is unlikely, as intellectual freedom was a core element of the Aufklarung and was something cherished by many intellectuals in this period.
Finally, the author equates Wiedergeburt with conversion. There are certainly similarities, but these are two different concepts and should not be confused.
The above issues suggestion that there is still more work to be done on this article and that work won't be easy. But it is worth doing, I think, and the result will be a much more robust and impactful piece of scholarship.
Author Response
I thank the reviewers for their insights and helpful comments. I've attached the updated version of the article. I've added 2000 words on the historiography and the historical-theoretical aspects of German nationalism. I've also updated the thesis to reflect these changes.

Round 2
Reviewer 2 Report
Comments and Suggestions for Authors
This article is significantly improved. There is still more work to be done, especially on the historiography, but I can see the light at the end of the tunnel. Please see my marginal comments on the attached pdf for details.

Author Response
Thank you for your thoughtful and specific comments for revision. I have addressed all the comments in my revised article. I've paid particular attention to the historiography of Schleiermacher's contribution to German nationalism. I've also completely revised the abstract. Let me know if you have any further questions or comments for revision. Thank you for helping to make this article a stronger contribution to the field.